# Interpolation Technique to Speed Up Gradients Propagation in Neural ODEs

**Talgat Daulbaev, Alexandr Katrutsa, Larisa Markeeva, Julia Gusak,**
**Andrzej Cichocki, Ivan Oseledets**
Skolkovo Institute of Science and Technology
Moscow, Russia
`talgat.daulbaev@skoltech.ru`
`aleksandr.katrutsa@phystech.edu`
`{l.markeeva,y.gusak,a.cichocki,i.oseledets}@skoltech.ru`

## Abstract

We propose a simple interpolation-based method for the efficient approximation of gradients in neural ODE models. We compare it with the reverse dynamic method (known in the literature as "adjoint method") to train neural ODEs on classification, density estimation, and inference approximation tasks. We also propose a theoretical justification of our approach using logarithmic norm formalism. As a result, our method allows faster model training than the reverse dynamic method that was confirmed and validated by extensive numerical experiments for several standard benchmarks.

## 1 Introduction

We propose a novel method to train neural ordinary differential equations (neural ODEs) [1]. This method performs stable and memory-efficient backpropagation through the solution of initial value problems (IVP). Throughout the study, we use the term neural ODEs for all neural networks with so-called ODE blocks. An ODE block is a continuous analog of a residual neural network [2] that can be considered as Euler discretization of ordinary differential equations. Neural ODEs have already been successfully applied to various machine learning problems, including classification, generative modeling, and time series prediction [1, 3, 4].

ODE block is a neural network layer that takes the activations $z_0$ from the previous layer as input, solves the initial value problem (IVP) described as

$$\begin{cases} \frac{\mathrm{d}z}{\mathrm{d}t} = f(z(t), t, \theta), & t \in [t_0, t_1] \\ z(t_0) = z_0, \end{cases} \tag{1}$$

and solved by any ODE solver. Note, the right-hand side in IVP (1) depends on set of parameters $\theta$ which is gradually updated during training. Therefore, during the backward pass in neural ODEs, the loss function $L$, which depends on the solution of the IVP, should be differentiated with respect to parameter $\theta$. A direct application of backpropagation to ODE solvers require a huge memory, since for every time step $\tau_k$, the output $z(\tau_k)$ must be stored as a part of the computational graph.

However, the approach based on the *adjoint method* [5, 6] helps to propagate gradients through the initial value problem with a relatively small memory footprint. Originally adjoint method has been actively used in mathematical modeling, for example, in seismograph and climate studies [7, 8]. It was used to investigate the sensitivity of the model output with respect to the input. In the context of training neural ODEs, the adjoint method is modified to combine it with the standard backpropagation [1]. We refer to this modified adjoint method as *reverse dynamic method* (RDM).

This method yields solving the augmented initial value problem backward-in-time. We call this IVP as the *adjoint IVP*. One of the components of the adjoint IVP is IVP that defines $z(t), t \in [0, 1]$. Therefore, the numerical solving of the adjoint IVP does not require storing intermediate activations $z(\tau_k)$ during the forward pass. As a result, the memory footprint becomes smaller.

However, Gholami et al. [9] showed that the RDM might lead to catastrophic numerical instabilities. To address this issue, the authors introduce a method called ANODE. This method exploits *checkpointing* idea [10], i.e., propose to store checkpoints $z(\tau_k)$ at intermediate selected time points $\tau_k$ during the forward pass. As result of this, in the backward pass, ANODE performs additional ODE solver steps forward-in-time in each interval between sequential checkpoints. The intermediate activations are stored to compute the target gradient. The main disadvantage of ANODE is that it requires intermediate activations storage and need to perform additional ODE solver steps.

To address the instability of the reverse dynamic method and limitations of ANODE, we propose *interpolated reverse dynamic method* (IRDM), which is described in detail in Section 2. This method is based on a smooth function interpolation technique to approximate $z(t)$ and exclude the IVP that defines $z(t)$ from the adjoint IVP. Thus, we do not reverse IVP (1) and avoid the instability problem of the reverse dynamic method. Under mild conditions on the right-hand side $f(z(t), t, \boldsymbol{\theta})$, function $z(t)$ is continuously differentiable as a solution of IVP, i.e. $z(t) \in C^1[t_0, t_1]$. Therefore, it can be approximated with the *barycentric Lagrange interpolation* (BLI) [11] on a Chebyshev grid. This technique is widely used for interpolation problems [11]. To construct such approximation, one has to store activations $z(t)$ in the point from the Chebyshev grid during the forward pass. These activations can be computed with DOPRI5 adaptive ODE solver without additional right-hand side evaluations [12]. After that, in the backward pass, stored activations are used to approximate $z(t)$ during the adjoint IVP solving. The main requirement for our method to work correctly and efficiently is that $z(t)$ can be approximated by BLI with sufficient accuracy. This can only be verified experimentally. However, the accuracy of such approximation is inherently related to the smoothness of the solution, which is also one of the main motivations behind using neural ODEs.

Our main contributions are the following.

- We propose *interpolated reverse dynamic method* to train neural ODEs. This method uses approximated activations $z(t)$ in the backward pass and reduces the dimension of the initial value problem, that is used to compute the gradient. Thus, the training becomes faster.

- We present the error bound for the gradient norm under small perturbation of the activations $z(t)$ induced by using interpolated values.

- We have evaluated our approach on density estimation, inference approximation, and classification tasks and showed its effectiveness in terms of test loss-training time trade-off compared to the reverse dynamic method.

## 2 Interpolated Reverse Dynamic Method

Deep learning problems are usually solved by minimizing a loss function $L$ with respect to model parameters using gradient-based methods. To compute the gradient $\frac{\partial L}{\partial \boldsymbol{\theta}}$ without saving computational graph from the forward pass, the *adjoint method* can be used [13, 14].

**Adjoint method.** The main idea of the adjont method is to derive gradients of the loss function $L$ from the first-order optimality conditions (FOOC) for a constrained optimization problem. In our case, the optimization problem is formulated as the loss minimization with ODE constraint in the form of (1).

To construct the corresponding Lagrangian, the adjoint variable $a(t)$ is introduced

$$\mathcal{L}(z(t), \boldsymbol{\theta}, a(t)) = L(z(t_1), \boldsymbol{\theta}) + \int_{t_0}^{t_1} a(t) \left( \frac{\mathrm{d}z}{\mathrm{d}t} - f(z(t), t, \boldsymbol{\theta}) \right) dt,$$

and the FOOC can be written in the following form

$$\frac{\delta \mathcal{L}}{\delta \boldsymbol{a}(t)} = \boldsymbol{0} \rightarrow \frac{\mathrm{d}\boldsymbol{z}}{\mathrm{d}t} - f(\boldsymbol{z}(t), t, \boldsymbol{\theta}) = \boldsymbol{0} \tag{2}$$

$$\frac{\delta \mathcal{L}}{\delta \boldsymbol{z}(t)} = \boldsymbol{0} \rightarrow \begin{cases} \frac{\mathrm{d}\boldsymbol{a}}{\mathrm{d}t} = -\boldsymbol{a}(t)^\top \frac{\partial f(\boldsymbol{z}(t), t, \boldsymbol{\theta})}{\partial \boldsymbol{z}} \\ \boldsymbol{a}(t_1) - \frac{\partial L}{\partial \boldsymbol{z}(t_1)} = \boldsymbol{0} \end{cases} \tag{3}$$

$$\frac{\delta \mathcal{L}}{\delta \boldsymbol{\theta}} = \boldsymbol{0} \rightarrow \frac{\partial L}{\partial \boldsymbol{\theta}} = \int_{t_0}^{t_1} \boldsymbol{a}(t)^\top \frac{\partial f(\boldsymbol{z}(t), t, \boldsymbol{\theta})}{\partial \boldsymbol{\theta}} dt. \tag{4}$$

Hence, the target gradient $\frac{\partial L}{\partial \boldsymbol{\theta}}$ can be computed in the following way: ODE (2) gives the activation dynamic $\boldsymbol{z}(t)$, ODE (3) gives the adjoint variable $\boldsymbol{a}(t)$ based on $\boldsymbol{z}(t)$ and finally the target gradient $\frac{\partial L}{\partial \boldsymbol{\theta}}$ is computed with the integral in (4). The adjoint method assumes that activations $\boldsymbol{z}(t_0) = \boldsymbol{z}_0$ are known in the backward pass. Thus, IVP (1) is solved forward-in-time, IVP (3) is solved backward-in-time and integral (4) is computed based on the derived $\boldsymbol{a}(t)$ and $\boldsymbol{z}(t)$. The adjoint method requires storing gradients $\frac{\partial f}{\partial \boldsymbol{\theta}}$ and $\frac{\partial f}{\partial \boldsymbol{z}}$ in intermediate activations $\boldsymbol{z}(t), t \in [t_0, t_1]$. Therefore, to reduce its memory consumption, the checkpointing idea is used.

**Checkpointing in the adjoint method.** ANODE method [9] exploits checkpointing idea to get the target gradient $\frac{\partial L}{\partial \boldsymbol{\theta}}$. This method stores some intermediate activations $\boldsymbol{z}(t)$ in the forward pass. These activations are called checkpoints. In the backward pass, ANODE considers intervals between sequential checkpoints from the right side to the left side. In every interval, ODE (2) with an initial condition equal to the checkpoint on the left is solved forward-in-time, IVP (3) is solved backward-in-time and the integral is updated. This approach is illustrated in Figure 1d. This approach still requires additional memory to store checkpoints and gradients $\frac{\partial f}{\partial \boldsymbol{\theta}}$. Also, it solves ODE (2) with multiple initial conditions equal to checkpoints. These drawbacks are fixed in reverse dynamic method [1].

**Reverse dynamic method.** This method is used in [1], where the neural ODE model is proposed, under the name "adjoint method". The reverse dynamic method assumes that activations $\boldsymbol{z}(t_1) = \boldsymbol{z}_1$ are known and do not store any checkpoints during the backward pass. Therefore, in the backward pass, the following IVP is solved backward-in-time and defines activaions:

$$\begin{cases} \frac{\mathrm{d}\boldsymbol{z}}{\mathrm{d}t} = f(\boldsymbol{z}(t), t, \boldsymbol{\theta}) \\ \boldsymbol{z}(t_1) = \boldsymbol{z}_1, \end{cases} \tag{5}$$

IVP (3) is solved backward-in-time and integral (4) is computed as the solution of the following IVP:

$$\begin{cases} \frac{\mathrm{d}}{\mathrm{d}t} \left( \frac{\partial L}{\partial \boldsymbol{\theta}} \right) = -\boldsymbol{a}(t)^\top \frac{\partial f(\boldsymbol{z}(t), t, \boldsymbol{\theta})}{\partial \boldsymbol{\theta}} \\ \frac{\partial L}{\partial \boldsymbol{\theta}}(t_1) = \boldsymbol{0}. \end{cases} \tag{6}$$

Thus, IVPs (3),(5) and (6) can be composed in the augmented IVP that is solved backward-in-time. This method is illustrated in Figure 1a. The study [9] demonstrates that this method can be unstable due to the reverse IVP (1). To get the right trade-off between stability and memory consumption, we propose *interpolated reverse dynamic method* (IRDM).

**Interpolated reverse dynamic method.** In the proposed *interpolated reverse dynamic method* (IRDM), we suggest to eliminate (5) from the adjoint IVP. Instead of using IVP (5) to get activations $\boldsymbol{z}(t)$, the IRDM approximates them through the barycentric Lagrange interpolation (BLI) on a Chebyshev grid [11]. This method is summarized in Figure 1c. We urge readers not to confuse the Lagrange interpolation, which is mostly of theoretical interest, with the BLI, that is widely used in practice for polynomial interpolation [15].

Denote by $\hat{\boldsymbol{z}}(t)$ the interpolated activations with the BLI technique that are used in the backward pass. As described in [15], $\hat{\boldsymbol{z}}(t)$ can be computed with the following equation:

$$\hat{\boldsymbol{z}}(t) = \left( \sum_{n=0}^{N} \frac{w_n}{t - \tau_n} \hat{\boldsymbol{z}}_n \right) \Big/ \left( \sum_{n=0}^{N} \frac{w_n}{t - \tau_n} \right), \tag{7}$$

where the sequence $\{\tau_n\}_{n=0}^N$ form the Chebyshev grid, and $t_0 = \tau_0 < \tau_1 < \ldots < \tau_N = t_1$, $\hat{\boldsymbol{z}}_n \triangleq \boldsymbol{z}(\tau_n)$ are exact activations computed in the Chebyshev grid during the forward pass and stored

to be used in the backward pass. To get these activations during the forward pass, we explore features of DOPRI5 adaptive solver [12] to compute activations in given time points (e.g., in Chebyshev grid) without additional right-hand side evaluations. Thus, we store $z(\tau_n)$ and solve IVP (1) simultaneously. The weights $w_n$ are computed as follows once for the entire training process

$$w_n = (-1)^n \sin\left(\frac{(2n+1)\pi}{2N+2}\right).$$

The computational complexity of computing $\hat{z}(t)$, as well as additional memory usage, is $O(N)$. Since we approximate $z(t)$, only (3) and (6) have to be solved backward-in-time during the backward pass. Thus, the dimension of the backward IVP is reduced by the size of the activations $z(t)$.

From the theory of polynomial interpolation, it is known, that if $z(t)$ is analytic function, then the following bound on the BLI approximation error holds

$$\max_{t\in[0,1]} \|\hat{z}(t) - z(t)\|_\infty \leq \mathcal{O}(M^{-N}), \tag{8}$$

where $M > 1$ depends on the region where the activation dynamic $z(t)$ is analytic, more details see in [15, 16, 17]. The solution of IVP (1) is analytic if the right-hand side $f(z(t), t, \theta)$ is analytic [18].

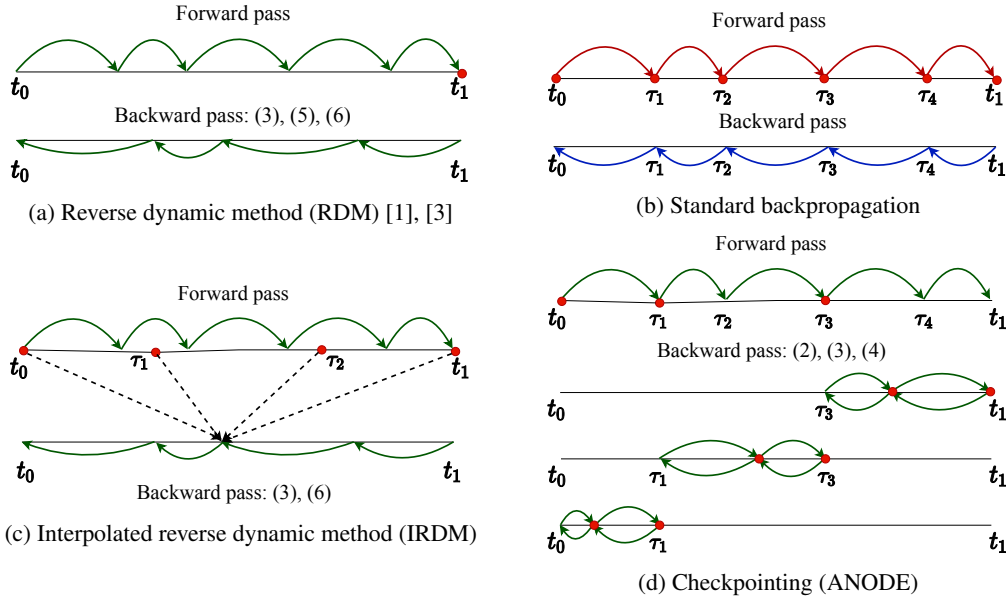

Figure 1: Comparison of different schemes to make forward and backward passes through the ODE block. Red circles indicate that the activations are stored at these time points. Red arrows indicate that during ODE steps, the outputs of the intermediate layer are stored to propagate gradients. Green arrows correspond to the steps with ODE solvers. Blue arrows correspond to the steps with automatic differentiation through the stored computational graph. Activations in Chebyshev grid points ($t_0, \tau_1, \tau_2$ and $t_1$ in Figure 1c) are stored in the interpolation approach during the forward pass. Chebyshev grid points do not necessarily coincide with time steps of ODE solver, but activations in these points can be recovered from the computed activations with ODE solver. The stored activations are used to approximate activations in the backward pass. The dotted arrows in Figure 1c shows that activations in $t_0, \tau_1, \tau_2$ and $t_1$ are used to interpolate activations in the backward pass.

## 3 Upper Bound on the Gradient Error Induced by Interpolated Activations

The proposed method excludes IVP that defines $z(t)$ from the adjoint IVP and uses approximation $\hat{z}(t)$ given by the barycentric Lagrange interpolation formula (7). Therefore, the dimension of the adjoint IVP is reduced, but the error in gradient $\frac{\partial L}{\partial \theta}$ appears since the activations are not exact. In this section, we present the upper bound on the gradient error norm and the factors that affect the

magnitude of this error. By the gradient error norm, we mean the norm of the difference between gradients computed with exact activations $z(t)$ and interpolated ones $\hat{z}(t)$.

According to (4) we have to estimate the following error norm, where the 2-norm is used

$$E = \left\| \int_{t_0}^{t_1} \left[ \tilde{a}(t)^\top \frac{\partial f(\tilde{z}(t), t, \boldsymbol{\theta})}{\partial \boldsymbol{\theta}} - \boldsymbol{a}(t)^\top \frac{\partial f(\boldsymbol{z}(t), t, \boldsymbol{\theta})}{\partial \boldsymbol{\theta}} \right] dt \right\|, \tag{9}$$

where $\tilde{a}(t)$ and $\tilde{z}(t)$ are adjoint variables and activations obtained with interpolation technique and $\boldsymbol{a}(t)$ and $\boldsymbol{z}(t)$ are exact ones. Now we derive the upper bound estimate of $E$ and show what factors affect this upper bound. Introducing perturbations $\Delta \boldsymbol{z}(t)$ and $\Delta \boldsymbol{a}(t)$ such that $\tilde{a}(t) = \boldsymbol{a}(t) + \Delta \boldsymbol{a}(t)$ and $\tilde{z}(t) = \boldsymbol{z}(t) + \Delta \boldsymbol{z}(t)$ and using the first-order expansion of $\frac{\partial f}{\partial \boldsymbol{\theta}}$ at $\boldsymbol{z}(t)$ we can re-write terms from (9) with perturbed adjoint variables and activations as follows

$$\tilde{a}(t)^\top \frac{\partial f(\tilde{z}(t), t, \boldsymbol{\theta})}{\partial \boldsymbol{\theta}} = (\boldsymbol{a}(t) + \Delta \boldsymbol{a}(t))^\top \left( \frac{\partial f(\boldsymbol{z}(t), t, \boldsymbol{\theta})}{\partial \boldsymbol{\theta}} + \frac{\partial^2 f(\boldsymbol{z}(t), t, \boldsymbol{\theta})}{\partial \boldsymbol{\theta} \partial \boldsymbol{z}} \Delta \boldsymbol{z}(t) + \mathcal{O}(\|\Delta \boldsymbol{z}(t)\|^2) \right)$$

Substitution this equality in (9) and applying standard inequalities lead to the following upper bound

$$E \le \int_{t_0}^{t_1} \|\boldsymbol{a}(t)\| \|\Delta \boldsymbol{z}(t)\| \left\| \frac{\partial^2 f(\boldsymbol{z}(t), t, \boldsymbol{\theta})}{\partial \boldsymbol{\theta} \partial \boldsymbol{z}} \right\| dt + \int_{t_0}^{t_1} \|\boldsymbol{a}(t)\| \left\| \frac{\partial f(\boldsymbol{z}(t), t, \boldsymbol{\theta})}{\partial \boldsymbol{\theta}} \right\| dt +$$
$$\int_{t_0}^{t_1} \|\Delta \boldsymbol{a}(t)\| \|\Delta \boldsymbol{z}(t)\| \left\| \frac{\partial^2 f(\boldsymbol{z}(t), t, \boldsymbol{\theta})}{\partial \boldsymbol{\theta} \partial \boldsymbol{z}} \right\| dt + M_{\boldsymbol{z}} \int_{t_0}^{t_1} (\|\boldsymbol{a}(t)\| + \|\Delta \boldsymbol{a}(t)\|) \|\Delta \boldsymbol{z}(t)\|^2 dt, \tag{10}$$

where $M_{\boldsymbol{z}} > 0$ is a constant hidden in big-O notation. In order to estimate the error norm, we have to analyze bounds for all terms in (10). The norm of activations perturbation $\|\Delta \boldsymbol{z}(t)\|$ is bounded according to the upper bound on the interpolation error (8). At the same time, the gradient $\frac{\partial f(\boldsymbol{z}(t), t, \boldsymbol{\theta})}{\partial \boldsymbol{\theta}}$ and the second partial derivative $\frac{\partial^2 f(\boldsymbol{z}(t), t, \boldsymbol{\theta})}{\partial \boldsymbol{\theta} \partial \boldsymbol{z}}$ are not bounded a priori, so we need to consider them additionally. The remaining terms are $\|\boldsymbol{a}(t)\|$ and $\|\Delta \boldsymbol{a}(t)\|$ and to bound them we need the following Lemma [19].

**Lemma 3.1** *([19], see Lemma 1)*

*Let $\boldsymbol{x}(t)$ be a solution of the following non-autonomous linear system*

$$\begin{cases} \dfrac{\mathrm{d}\boldsymbol{x}}{\mathrm{d}t} = \boldsymbol{x}(t)^\top \boldsymbol{A}(t) + \boldsymbol{b}(t), \\ \boldsymbol{x}(t_0) = \boldsymbol{x}_0. \end{cases} \tag{11}$$

*Then $\|\boldsymbol{x}(t)\| \le \xi(t)$, where the scalar function $\xi$ satisfies the IVP*

$$\begin{cases} \dfrac{\mathrm{d}\xi}{\mathrm{d}t} = \mu[\boldsymbol{A}(t)]\xi + \|\boldsymbol{b}(t)\|, \\ \xi(t_0) = \|\boldsymbol{x}_0\|, \end{cases} \tag{12}$$

*where $\mu[\boldsymbol{A}] \triangleq \lim\limits_{h \to 0+} \dfrac{\|\boldsymbol{I} + h\boldsymbol{A}\| - 1}{h}$ is a logarithmic norm of matrix $\boldsymbol{A}$ [20].*

If the 2-norm is used in definition of the logarithmic norm, then

$$\mu[\boldsymbol{A}] = \lambda_{\max} \left( \frac{\boldsymbol{A} + \boldsymbol{A}^\top}{2} \right),$$

where $\lambda_{\max}(\boldsymbol{A})$ is the maximum eigenvalue of a matrix $\boldsymbol{A}$.

In our case, IVP (11) is equivalent to IVP (3). Therefore, this lemma helps to estimate $\|\boldsymbol{a}(t)\|$. In particular, $\|\boldsymbol{a}(t)\| \le \xi(t)$, where $\xi(t)$ is a solution of the following IVP:

$$\begin{cases} \dfrac{\mathrm{d}\xi}{\mathrm{d}t} = \mu[\boldsymbol{J}(t)]\xi, \\ \xi(t_1) = \left\| \frac{\partial L}{\partial \boldsymbol{z}_1} \right\|, \end{cases} \tag{13}$$

where $\boldsymbol{J}(t) \triangleq \frac{\partial f(\boldsymbol{z}(t),t,\boldsymbol{\theta})}{\partial \boldsymbol{\theta}}$. Hence, the upper bound on the adjoint variable norm is written with the solution of IVP (13):

$$\|\boldsymbol{a}(t)\| \leq \xi(t_1) \exp \left( \int_{t_1}^{t} \mu[\boldsymbol{J}(\tau)]d\tau \right).$$

(14)

The upper bound for $\|\Delta\boldsymbol{a}(t)\|$ can be also obtained with Lemma 3.1. To derive this upper bound, we compose an auxiliary IVP that defines a dynamic of $\Delta\boldsymbol{a}(t)$. Consider the following IVPs corresponding to exact and perturbed activations:

$$\begin{cases} \frac{d\boldsymbol{a}}{dt} = \boldsymbol{a}(t)^\top \frac{\partial f(\boldsymbol{z}(t),t,\boldsymbol{\theta})}{\partial \boldsymbol{z}} \\ \boldsymbol{a}(t_1) = \frac{\partial L}{\partial \boldsymbol{z}(t_1)} \end{cases} \quad \begin{cases} \frac{d\tilde{\boldsymbol{a}}}{dt} = \tilde{\boldsymbol{a}}(t)^\top \frac{\partial f(\tilde{\boldsymbol{z}}(t),t,\boldsymbol{\theta})}{\partial \boldsymbol{z}} \\ \tilde{\boldsymbol{a}}(t_1) = \frac{\partial L}{\partial \boldsymbol{z}(t_1)}. \end{cases}$$

(15)

Subtracting one IVP from the other, we get the IVP that defines dynamic of $\Delta\boldsymbol{a}(t) = \tilde{\boldsymbol{a}}(t) - \boldsymbol{a}(t)$:

$$\begin{cases} \frac{d\Delta\boldsymbol{a}(t)}{dt} = \Delta\boldsymbol{a}(t)^\top \boldsymbol{J}(t) + \tilde{\boldsymbol{a}}(t)(\tilde{\boldsymbol{J}}(t) - \boldsymbol{J}(t)) \\ \Delta\boldsymbol{a}(t_1) = \boldsymbol{0}. \end{cases}$$

(16)

Note that IVP (16) satisfies assumption in Lemma 3.1. Therefore, the following estimate holds

$$\|\Delta\boldsymbol{a}(t)\| \leq \xi(t),$$

(17)

where $\xi(t)$ is a solution of the following IVP:

$$\begin{cases} \frac{d\xi}{dt} = \mu[\boldsymbol{J}(t)]\xi + \|\tilde{\boldsymbol{a}}(t)(\tilde{\boldsymbol{J}}(t) - \boldsymbol{J}(t))\|, \\ \xi(t_1) = \boldsymbol{0}. \end{cases}$$

(18)

The solution of IVP (18) is given by the following formula

$$\xi(t) = \phi(t) \int_{t_1}^{t} \phi^{-1}(\tau)\|\tilde{\boldsymbol{a}}(\tau)(\tilde{\boldsymbol{J}}(t) - \boldsymbol{J}(t))\|d\tau,$$

(19)

where $\phi(t) = \exp \left( \int_{t_1}^{t} \mu[\boldsymbol{J}(\tau)]d\tau \right)$ is a fundamental solution of IVP (19). Thus, we get the upper bounds for all terms in the inequality (10).

Thus, we can list the factors that affect the accuracy of gradient approximation with interpolated activations. These factors are constants that bound $\frac{\partial f(\boldsymbol{z}(t),t,\boldsymbol{\theta})}{\partial \boldsymbol{\theta}}$ and $\frac{\partial^2 f(\boldsymbol{z}(t),t,\boldsymbol{\theta})}{\partial \boldsymbol{\theta} \partial \boldsymbol{z}}$ for $t \in [t_0, t_1]$ (10), logarithmic norm $\mu[\boldsymbol{J}(t)]$ (14), (17), (19), and smoothness of $\boldsymbol{J}(t)$ (19).

## 4 Numerical Experiments

In this section, firstly, we compare the proposed the IRDM with the RDM on the density estimation and variational inference tasks (for the RDM baselines, we use FFJORD [3] implementation). Secondly, we show the benefits of the IRDM on the CIFAR10 classification task (RDM implementation is similar to [1]). We demonstrate that during training, the IRMD requires less computational time to achieve convergence and a smaller number of evaluations of the right-hand side function comparing to the baselines. The source code of the proposed method can be found at GitHub[1].

Also, as the number of Chebyshev grid points $N$ is an important hyperparameter in our method, we study how it affects the gain in considered tasks. Our method is implemented on top of `torchdiffeq`[2] package. The default ODE solver in our experiments is the DOPRI5. The values of optimized hyperparameters are in the supplementary materials. Mostly we follow the strategies from [3] and [9]. Every separate experiment is conducted on a single NVIDIA Tesla V100 GPU with 16Gb of memory [21]. We conducted all experiments with three different fixed random seeds and reported the mean value. Experiments were tracked using the "Weights & Biases" library [22].

**Improvement in stability of gradient computations.** We perform experiments on the reconstruction trajectory of the dynamical system that collapses in zero. As a result, we observe that the reverse dynamic method (RDM) and our method (IRDM) solve this problem with approximately the same accuracy. However, the RDM requires at least 10 times more right-hand side evaluations to solve adjoint IVP in every iteration than the IRDM. We use RDM implementation from the `torchdiffeq` package. Thus, in such a toy problem the IRDM and the RDM compute similar gradients, but the IRDM computes them much faster. To illustrate the stability of the IRDM, we show the plot of test loss vs. training time in density estimation problem, see Figure 3a.

**How gradient approximation depends on the tolerance in adaptive solver and size of the Chebyshev grid.** In Section 3, we provide theoretical bounds on the gradient approximation and list the main factors that affect it. However, tolerance in the adaptive solver and number of nodes in the Chebyshev grid can also significantly affect the quality of gradient approximation. To illustrate this influence empirically, we consider the toy dynamical system in 2D with the right-hand side $Ay^3$ and train neural ODE model to approximate trajectories of this dynamical system. We consider the range of tolerances in the DOPRI5 adaptive solver and the range of nodes quantities in the Chebyshev grid. The result of this experiment is presented in Figure 2. This plot shows that the smaller tolerance, the more accurate gradients approximation for all considered number of nodes in the Chebyshev grid. At the same time, the larger number of nodes leads to decreasing the approximation quality.

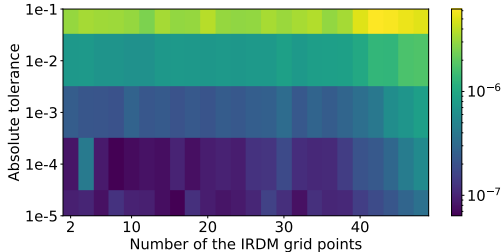

Figure 2: The dependence of the IRDM gradients error in $\ell_1$-norm with respect to the number of nodes in the Chebyshev grid and the tolerance of the DOPRI5 method. The output of the standard backpropagation performed for the DOPRI5 with `1e-7` tolerance was used as a ground truth.

## 4.1 Density Estimation

The problem of density estimation is to reconstruct the probability density function using a set of given data points. We compared the proposed IRDM with the RDM (FFJORD[3] baseline) that exploits the reverse dynamic method to density estimation problem. We tested these methods on four toy datasets (`2spirals`, `pinwheel`, `moons` and `circles`) and tabular `miniboone` dataset [23]. This tabular dataset was used in our experiments since it is large enough and allows considered methods to converge for a reasonable time. According to [3] setting, we stopped the training process if, for the sequential 30 epochs, the test loss does not decrease. Therefore, we excluded test loss values given by the last 30 epochs from the plots. The model for `miniboone` was slightly different from the model from [3]; it includes 10 ODE blocks instead of 1. We used Adam optimizer [24] in all tests on the density estimation problem. For toy datasets, we used the following hyperparameters: learning rate equals $10^{-3}$, the number of epochs was 10000, the batch size was 100, absolute and relative tolerances in the DOPRI5 solver were $10^{-5}$ and $10^{-5}$.

Figure 3 shows that the test loss decreases more rapidly with our method than with the RDM. To make figures more clear, we plot convergence only for one value of $N$ for every dataset. This value of $N$ gives the best result among the tested values. Similar graphs for toy datasets can be found in Supplementary materials.

The number of nodes in the Chebyshev grid significantly affects the performance of the proposed method. If this number is small, then the interpolation accuracy is not enough, and the training takes a long time. If this number is large, then the computing of intermediate activations is too costly, and training is relatively slow. In supplementary materials, we provide graphs with empirical results on

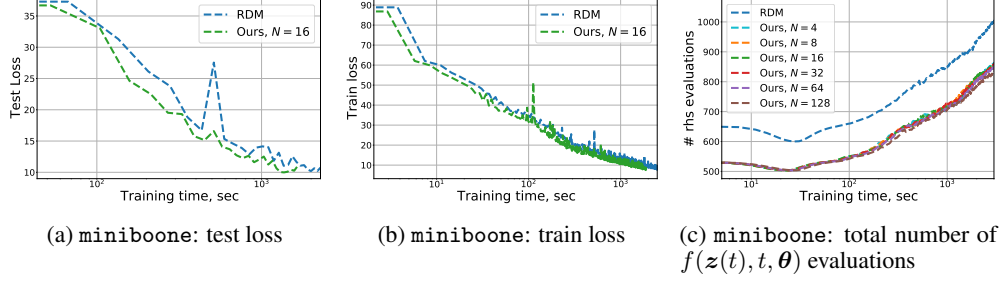

(a) `miniboone`: test loss     (b) `miniboone`: train loss     (c) `miniboone`: total number of $f(\boldsymbol{z}(t), t, \boldsymbol{\theta})$ evaluations

Figure 3: Comparison of the IRDM with the RDM (baseline from FFJORD) on density estimation problem for tabular dataset `miniboone`. The number of points in the Chebyshev grid $N$ used in the IRDM is given in the legend.

how the number of points in the Chebyshev grid affects the convergence rate. On Figure 4, the total number of the right-hand side $f(\boldsymbol{z}(t), t, \boldsymbol{\theta})$ evaluations per training iteration is shown.

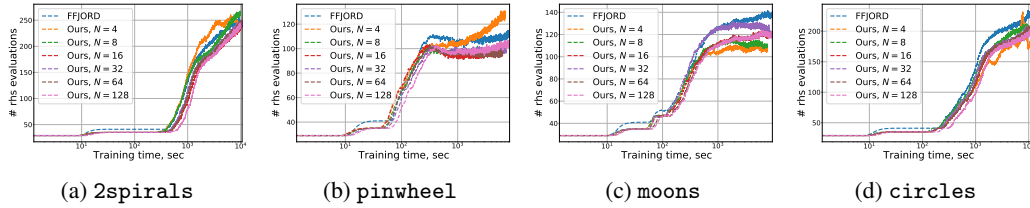

(a) `2spirals`     (b) `pinwheel`     (c) `moons`     (d) `circles`

Figure 4: Total number of $f(\boldsymbol{z}(t), t, \boldsymbol{\theta})$ evaluations for density estimation datasets.

## 4.2 Variational Autoencoder

We also compare the RDM (baselines from FFJORD) and the IRDM on the training variational autoencoder [25]. We use datasets `caltech` and `freyfaces`. The test negative ELBO loss and test bits per dim loss are reported for `caltech` and `freyfaces` datasets, respectively. Figure 5 illustrates a minor acceleration of convergence provided by the IRDM compared to the RDM. However, the IRDM gives the same final test loss with the same training time as the RDM. We suppose that the reason for such convergence degradation near the optimum is the same as for the density estimation models.

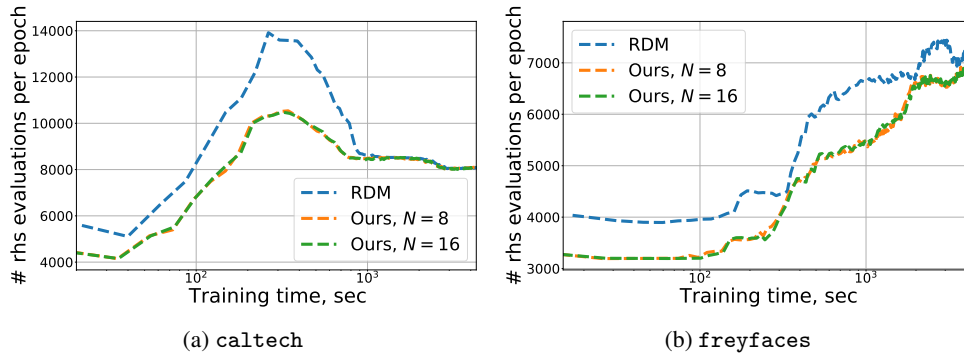

(a) `caltech`              (b) `freyfaces`

Figure 5: Comparison of the number of right-hand side evaluations for the IRDM and the RDM in training variational autoencoder.

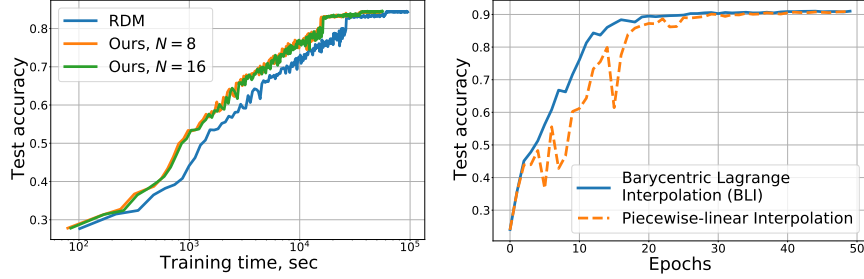

(a) Comparison of the IRDM with the RDM in CIFAR10 classification task. IRDM even with $N = 8$ nodes trains faster than RDM.

(b) Comparison of BLI and piecewise-linear interpolation used in the IRDM (8 nodes) in MNIST classification problem.

Figure 6: Experiments results in the image classification task. The reported values are averaged over three trained models corresponding to the considered tasks.

## 4.3 Classification

We test the proposed method on the classification problem with the CIFAR10 dataset. The model with a single convolution, a single ODE block, and a linear layer is considered. For this model, the IRDM with 16 points in the Chebyshev grid gives $0.867$ test accuracy with the batch size 512 and tolerance `1e-3` in the DOPRI5. We compare the IRDM with the RDM in terms of test loss versus training time. Figure 6a demonstrates that the IRDM gives higher test accuracy and requires less training time.

Another experiment is investigating whether other interpolation techniques can be used in the IRDM. We compare the Barycentric Lagrange Interpolation (BLI), which is a default method used in the IRDM, with the piecewise-linear interpolation. We perform 50 epochs in the MNIST classification problem with constant learning rate `1e-1` and without data augmentation. Figure 6b confirms our choice of BLI since already after 40 epochs piecewise-linear interpolation provides less stable test accuracy behaviour.

## 5 Related Work

Neural ODEs [1] is a model inspired by the connection between neural networks and dynamical systems [26, 27, 28, 29]. Neural ODEs and its modifications were used for various different applications [3, 4, 30, 31, 32]. Nguyen et al. [33] emphasized the importance of using adaptive solvers and introduce a procedure to learn their tolerances. Quaglino et al. [34] proposed to use of spectral element methods, where the dynamics are expressed as a truncated series of Legendre polynomials. Similar ways of using interpolation in the adjoint method are implemented in SUNDIALS [35].

The proposed method relies on the ability of Runge-Kutta (RK) methods to evaluate the trajectory in intermediate points with Hermite polynomial interpolation. We use this feature of RK methods to evaluate activations in the Chebyshev grid points. The works by L. F. Shampine [36, 37] study the error induced by this approach to evaluate activations in intermediate points. In addition, the stiffness of ODE is an important concept [38, 39] for stable and fast training of neural ODEs.

## 6 Conclusion

We have presented the interpolated reverse dynamic method (IRDM) to improve the original reverse dynamic method (RDM) for training neural ODE models. The main idea of IRMD is to reduce the number of ODEs solved during the backward pass by using interpolated values $z(t)$ rather than ones found from equation (5). Thus, the total number of right-hand side evaluations during training and convergence time decreases compared to the original reverse dynamic method. We have empirically demonstrated this behavior on density estimation, variational inference, and classification tasks. Also, we have derived a theoretical upper bound on the error in computed gradients induced by the interpolation. The influence of the tolerance in adaptive ODE solver and the number of nodes in the Chebyshev grid is also studied numerically.

## Broader Impact

We proposed a method for fast and stable Neural ODEs training. This method can be applied to any domain, where it is possible to use Neural ODEs. Since the IRDM reduces the time needed to train Neural ODEs, it has the potential to reduce the carbon footprint of building AI models.

## Acknowledgement

Section 2 is supported by Ministry of Education and Science of the Russian Federation grant 14.756.31.0001. Section 3 was funded by RFBR, project number 19-29-09085 MK. Section 4 was funded by RFBR, project number 20-31-90127. High-performance computations presented in the paper were carried out on Skoltech HPC cluster Zhores [21].

## Footnotes

[1]`https://github.com/Daulbaev/IRDM`

[2]`https://github.com/rtqichen/torchdiffeq/`

[3]`https://github.com/rtqichen/ffjord`

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
