[Supplementary Material]

# Appendix

## A Number of Chebyshev grid points

In this section, we study how the quality of a model depends on the number of Chebyshev grid points. To demonstrate this dependence, we perform experiments with a range of $N$ on toy two-dimensional datasets for the density estimation problem. Figures 7a and 7d show that if the number of nodes is too small, e.g., $N = 4$, the IRDM converges to the higher test loss. It means that the interpolation accuracy is not enough, and a larger number of points in the Chebyshev grid is needed. On the other hand, Figure 7 illustrates that if the number of nodes is too large, e.g., $N = 128$, the IRDM might be slower than the RDM. The reason is that a large number of nodes leads to the costly computations of the interpolated activations, see Equation (7) in the main text. Since the right-hand side function in the ODE block for toy datasets is easy to compute, the speedup effect is not much noticeable. However, the total number of the right-hand side function evaluations performed in IRDM is significantly smaller than in RDM, see Figure 8. Therefore, the more computationally expensive the right-hand side function is in the ODE block, the more significant gain one can get from using IRDM.

(a) 2spirals

(b) pinwheel

(c) moons

(d) circles

Figure 7: Comparison of IRDM (our method) and RDM on density estimation problem for toy datasets 2spirals, pinwheel, moons, and circles in terms of test loss versus wall-clock training time. Comparison results for every dataset are presented in the corresponding subplot. The number of points in Chebyshev grid $N$ used in the IRDM is given in legend.

Table 1 shows the total time to perform 10000 training iterations for considered toy datasets. It can be seen that the IRDM with $N = 16$ nodes always outperforms the RDM.

## B Experimental settings

To integrate ODE blocks in all experiments, we use the DOPRI5 ODE solver. We report mean test loss values and function evaluations values. These values are computed based on the ten runs with different fixed random seeds for toy datasets and three runs for other datasets.

(a) `2spirals`

(b) `pinwheel`

(c) `moons`

(d) `circles`

Figure 8: Comparison of IRDM (our method) and RDM on density estimation problem for toy datasets `2spirals`, `pinwheel`, `moons`, and `circles` in terms of total number of the right-hand side evaluations versus number of iterations. Comparison results for every dataset are presented in the corresponding subplot. The number of points in Chebyshev grid $N$ used in the IRDM is given in legend.

Table 1: Time (in seconds) to perform 10000 training iterations for toy datasets.

| # nodes / dataset | RDM | 4 | 8 | 16 | 32 | 64 | 128 |
|---|---|---|---|---|---|---|---|
| `pinwheel` | 5318 | 5389 | **4642** | 4750 | 4846 | 5152 | 5574 |
| `circles` | 7874 | 6927 | 6864 | **6842** | 7037 | 7578 | 8113 |
| `moons` | 7494 | **5618** | 6062 | 6471 | 6518 | 6720 | 7192 |
| `2spirals` | 9285 | 8998 | 9492 | **8938** | 8947 | 9267 | 9680 |

## B.1 Classification

We test considered methods of neural ODE model training in the CIFAR10 classification task. We consider a model with a single ODE block, which consists of two convolutional layers with 64 input and output channels, ReLU activations, and weight normalizations. A convolutional layer with three input channels and 64 output channels, a batch normalization layer, and ReLU activation precede the ODE block. We use only random crops and random flips for data augmentation.

The training is performed by SGD with momentum $0.9$. The weight decay is equal to `1e-5`; the learning rate is fixed and equal to `5e-3`, batch size is 100. Initial absolute and relative tolerances are set to `1e-5`. After the $n^{th}$ 150 epoch, these tolerances are decreased by 10.

## B.2 Density estimation

For the density estimation problem, we consider `miniboone` tabular dataset and four two-dimensional toy datasets. Data and preprocessing procedures are taken from `https://github.com/gpapamak/maf` and `https://github.com/rtqichen/ffjord`, respectively.

Instead of simple linear layers inside the ODE block, we use a so-called `concatsquash` linear layers, which are defined as follows

$$(\boldsymbol{W}\boldsymbol{z} + \boldsymbol{b}_1) \odot \sigma\left(t\boldsymbol{c} + \boldsymbol{b}_2\right) + t\boldsymbol{b}_3, \tag{20}$$

where $\boldsymbol{z}$ are input activations, $t$ stands for the time, $\boldsymbol{W}, \boldsymbol{c}, \boldsymbol{b}_1, \boldsymbol{b}_2, \boldsymbol{b}_3$ are trainable parameters, $\sigma$ is a sigmoid activation, and $\odot$ is an element-wise product.

For the `miniboone` dataset, we train a model with 10 ODE blocks and softplus nonlinearity. It is trained by Adam optimizer with a fixed learning rate equal to `1e-3`. The batch size is equal to 10000. Absolute and relative tolerances are set to `1e-8` and `1e-7`, respectively. The training terminates if test loss does not decrease during 30 sequential epochs.

In experiments with toy datasets, a model with a single ODE block is used. This ODE block consists of three `concatsquash` linear layers of size $2 \times 64$, $64 \times 64$, and $64 \times 2$. To train the Neural ODEs model, SGD with momentum $0.9$ and fixed learning rate `1e-3` is used. Absolute and relative tolerances are set to `1e-5`. The number of iterations in the training procedure is 10000, and 100 samples compose mini-batch.

## B.3 VAE

For VAE experiments, we choose two datasets: `caltech` and `freyfaces`. Both datasets can be found in `https://github.com/riannevdberg/sylvester-flows`. All the experimental settings are exactly the same as in the FFJORD paper. The only difference is that IRDM is used to train models instead of RDM.