[Reviews · NeurIPS 2020]

Review 1

Summary and Contributions: In this paper, the authors propose an alternative to the adjoint method that uses the barycentric Lagrange interpolation to approximate intermediate activations. The authors also give a theoretical analysis of the gradient error bound. The proposed method is applied to density estimation and classification tasks.

Strengths: The slowness has been a major road blocker for wider applications of neural ODE. I think this paper is relevant to the community. The application of the barycentric Lagrange interpolation seems novel. The theoretical analysis also gives a guarantee of the error in the gradients.

Weaknesses: The experimental study seems weak. The datasets considered for the density estimation task are only synthetic datasets. If would be helpful to see the performance on at least one real data, such as MNIST.

Correctness: The theorectical conclusion seems correct. The the empirical methodology is correct.

Clarity: The paper is well motivated and clearly written.

Relation to Prior Work: The authors presented and discussed related methods in detail in Figure 1.

Reproducibility: Yes

Additional Feedback: Besides faster model training, are there other advantages of the proposed model compared to the adjoint method? My understanding is that by storing intermediate activations, the stability of backpropagation should also be improved. Are there any ways to demonstrate this, especially when the time span is long?


Review 2

Summary and Contributions: The paper discusses different computation strategies to compute the gradients in Neural ODE blocks and presents a method to better trade-off memory, speed and accuracy in the adjoint computation using Barycentric Lagrange Interpolation. A theorem regarding error bounds is presented and empirical analysis is included. As a heads-up to all involved parties: I have seen a similar submission during the ICML 2020 review process which I assume is an earlier version of this work.

Strengths: The problem is well motivated and clearly illustrated besides English style and grammar issues. I particularly commend Sect. 2 (with minor caveats detailed below) and Fig. 1. I agree with the authors that it is a very important problem in Neural ODEs which would be a strong argument for me to potentially accept the submission.

Weaknesses: I take issues with two aspects of this submission that lead me to recommend rejection at this point. 1.) The submission points out that the evaluations of z(t) at the Chebyshev grid points can be obtained without additional cost, e.g., at line 108. While this is true in some sense in general, there are many numerical theory aspects to this claim that are ignored here, both in the text as well as in the code. Runge-Kutta methods only guarantee high-order approximations at their own grid points. If high-order approximations are sought at pre-defined grid points, there are two solutions: a) the solvers are forced to include the pre-defined grid points as part of the otherwise adaptive mesh or b) a particular choice has to be made to find a smooth-interpolant Runge-Kutta formula. Neither solution is mentioned in the submission, nor applied in the code (I have checked), but smooth-interpolants exist for the DOPRI5 pair, e.g., https://www.sciencedirect.com/science/article/pii/0377042790901989 This in itself would not be a cruical omission. However, since the authors claim that the BLI in particular stands out as interpolation technique, this claim would need to be contrasted with other interpolation techniques to achieve similar results. 2.) There is little practical context for the bound of Sect. 3. All statements and proofs seem to be correct in Sect. 3 on my superficial reading, so this is not a problem. The problem is a) the proof compares to ground truth activations z(t) which are not available. At best we have the numerical solution of DOPRI5. If I understood the code correctly, the authors furthermore interpolate the DOPRI5 output to obtain the values at the Chebyshev grid which is then once more interpolated when needed during the adaptive step-size backward pass. All these error propagations are not considered in the presented theory. b) Even if this was presented, the user of Neural ODEs needs to know how the error compares to the reverse dynamic mode which is never directly evaluated in the manuscript, only indirectly through training time, #function evaluations and accuracy, which further includes the effects of stochastic optimization in the mix. Thus, the paper reserves two pages of theory that are fun to read, but not meaningful to the contributions of the paper.

Correctness: Claims seem to be correct. Empirical methodology is borderline, but would be acceptable. Hints to improve experiments: - comparison with ANODE could (should?) be included - Personally, I would like to see an evaluation of gradient accuracy with respect to grid points and tolerance for the RDM. To this end, I would like to see how gradient estimates converge when using lower and lower tolerances for the RDM and more and more grid points for the BLI. These convergence speeds could be contrasted with overall gradient variance through mini-batching. (If necessary, this could be done on synthetic data sets.) - An experiment should be conducted how much more function evaluations/wall clock time is needed to force DOPRI5 to include the Chebyshev grid vs. how the gradients differ if 'simple interpolation' is used vs. a RK smooth-interpolant. These experiments could take up the space of Sect. 3 and Sect. 3 could move in the appendix.

Clarity: Please have your submission proof-read for English style and grammar issues.

Relation to Prior Work: This is adequate.

Reproducibility: Yes

Additional Feedback: Since this is such an important problem and this really is standard numerical practice in other fields, I would be willing to be convinced that the paper should be accepted. To this end, I am looking for convincing answers to these questions in the rebuttal: a) Will the code be released upon acceptance? b) How do the authors plan to improve the manuscript with respect to weakness 1.)? Post-rebuttal update: I thank the authors for their feedback. I still maintain my position that the theorem does not support the flow of arguments of the main contribution. While the authors have suggested that based on these insights, regularizers could be formulated, this line of rationale is nowhere present in the rest of the manuscript. I think the clarity could further improved if this intention is clearly mentioned. I can follow their argument of the ANODE solver. I am looking forward to see the promised additional experiments with respect to solver tolerances, number of grid-points and piecewise-smooth interpolants. I am also looking forward to have a look at the code upon release. As a consequence of the rebuttal, I'll raise my score, but I'll lower my confidence, as I will only be able to see the promised additions conditioned on the acceptance decision.


Review 3

Summary and Contributions: This works tackles the problem of reversing a dynamical system during the adjoint method (or reverse dynamic method) for computing gradients of Neural ODEs. Instead of solving the state z(t) backwards in time, they propose replacing it with a barycentric Lagrange interpolation based on N points. These N points are computed during the forward pass for essentially free using an adaptive ODE solver and stored. The approach allows the user to change the memory required from O(1) to O(N) while increasing the accuracy of the state reconstruction. The authors also bound the error of this interpolation method, which grows wrt the gradient norms (Lipschitz constants) of f. Empirically, the proposed approach seems to be able to compute gradient marginally faster. -- Update -- I thank the authors for including an experiment on reversing a difficult system. I rescind my comment on the marginal time improvement, though it'd be best if the authors could include error bars.

Strengths: The method is simple and the code is provided.

Weaknesses: Though one motivation was being able to use this approach for ODE systems that are numerically unstable for the reverse dynamics method, this scenario was not tested. Notably, one aspect this paper has ignored is that the actual gradient vector (which is usually much larger compared to the state z(t)) still needs to be solved backwards in time using the reverse dynamic method, and it seems the computed gradients using both the reverse dynamic method and the integrated one are mostly the same, judging from the loss curves. So while the experiments show that there are fewer evaluations due to not having solve z(t) backwards, the paper would be stronger if it is actually applied to a system that the reverse dynamic method cannot solve. (For instance, an ODE that collapses to one point is easy to solve in the forward direction but difficult in the reverse direction.)

Correctness: I did not spot any mistakes.

Clarity: The structure and explanations in the paper are very well-written. The paper does have some small grammatical mistakes throughout. I would suggest the authors pass the text through a grammar checker.

Relation to Prior Work: A related work is SNODE: Spectral Discretization of Neural ODEs for System Identification [1] at ICLR 2020, where they use Legendre polynomials to approximate the ODE solution. [1] https://openreview.net/forum?id=Sye0XkBKvS

Reproducibility: Yes

Additional Feedback: The interpolation acts as an approximation for z(T). Is it possible to directly backpropagate through this approximated z(T) to obtain gradients with O(N) memory? If the interpolation sufficiently models the solution, this can forego solving the gradient vector.


Review 4

Summary and Contributions: This paper proposes an interpolation technique to speed up the approximation of gradients in ODEs via the adjoint method

Strengths: The authors present a bound on the error in their gradient approximation.

Weaknesses: The bound on the gradient error appears to not be particularly useful and rely on boundedness conditions on derivatives of the vector field, which are VERY strong.

Correctness: The overall methodology appears to be sound.

Clarity: The paper is mostly clearly written however at some places the word "the" appears to be missing, there"s some typos, abrupt changes in notation and unannounced notation overload.

Relation to Prior Work: Yes.

Reproducibility: Yes

Additional Feedback: Some comments: 1) The notation for the interpolant of z starts out being \hat{z} but then changes to \tilde{z}. 2) In the bound of E in Eq. (10) the second term does not have the approximation errors \Delta z or \Delta a as factors consequently, it can not be reduced by improving the approximation quality. This does not appear to be useful. 3) J starts being defined as df/d\theta but ends up being df/dz? ( Eq. (16)). 4) The boundedness condition on df/d\theta and d^2 f/d\theta dz is very strong. Can this even be verified in the considered applications?

[Author Response · NeurIPS 2020]

We want to thank the reviewers for their thorough comments and constructive suggestions! They highlighted the
importance of the formulated problem (**R1**, **R2**, **R3**) and the relevance of the submitted manuscript to the community
(**R1**). Also, reviewers noticed that the proposed method is simple (**R3**), the error bound on the gradient approximation is
derived (**R1**, **R4**), and the code is provided (**R3**). All reviewers commend the clear explanation but noted the language
issues and typos in formulas (**R4**). We respond to the reviewers' comments below and update the final version of the
manuscript.

**[R2, R4] Concerns about theoretical error bounds**. The error bound derivation was included to demonstrate which
factors affect the quality of gradient approximation. Namely, *logarithmic norm* of the right-hand side Jacobian affects
the approximation quality and can be further used as a regularizer when training neural ODE models.

**[R1, R3] Improvement in stability of gradient computations.** We perform additional experiments on reconstruction
trajectory of dynamical system that collapses in zero. As a result, we observe that reverse dynamic method (RDM)
and our method (IRDM) solve this problem with approximately the same accuracy. However, RDM requires at least
10 times more right-hand side evaluations to solve adjoint IVP in every iteration than IRDM. Thus, IRDM and RDM
compute similar gradients, but IRDM computes them much faster.

**[R1] Datasets considered for the density estimation task are only synthetic.** We present the performance of our
method for the density estimation problem on the real data `miniboone` dataset (Figure 2).

**[R2] How do the authors plan to improve the manuscript with respect to weakness 1?** We use a 4th order scheme
proposed in "Some Practical Runge-Kutta Formulas" by Shampine (1986). It combines the basic Dormand-Prince
Runge-Kutta process with their 5th-order formula, modified 4th-order formula, and 4th-order formula for the midpoint
along with local quartic Hermite interpolation. Theoretical results in "Interpolation for Runge-Kutta methods" by
Shampine (1985) yield that we get 4th-order approximations at pre-defined grid points. If we understand the terminology
right, we already use a smooth-interpolant RK formula. A useful paper by Calvo et. al., recommended by **R2**, can be
considered as a generalization of a 5th order scheme, proposed in the above paper by Shampine. We will include a
detailed description related to this issue and links to the mentioned papers in the final version.

Yes, a scheme in which the solvers are forced to include the predefined grid points as part of the otherwise adaptive
mesh, can be applied. However, such a scheme repeatedly increases the size of the optimal step size, which ruins the
control of optimal step size, and it can lead to a high increase in computational time and rounding error. We will add an
experiment to the final version to demonstrate our method's time savings compared to this approach.

**[R2] Comparison with ANODE.** ANODE exploits the checkpointing technique in the backward pass (see Figure 1d).
In the official repository, we found the implementation that exploits a standard automatic differentiation rather than
checkpointing. Therefore, we did not compare with ANODE since the out of memory error was raised during training.

**[R2] I would like to see an evaluation of gradient accuracy with respect to grid points and tolerance.** We carried
out additional experiments to estimate the requested dependencies and conclude that the larger (absolute and relative)
tolerances lead to larger error in gradient estimation. Also, the dependence of gradient accuracy w.r.t. number of grid
points is non-monotonic, i.e. too small and too large number of points in Chebyshev grid leads to degrading of gradient
accuracy. These results will be added to the final version of the manuscript.

**[R2] Will the code be released upon acceptance?** Sure, the code will be released.

**[R2] Barycentric Lagrange interpolation (BLI) should be contrasted with other interpolation techniques.** In-
deed, other interpolation techniques can be used in the proposed method. We tested piecewise-smooth interpolants
(such as piecewise-linear and cubic splines) and found that they provide worse performance than BLI. We will add
these experiments to the final version.

**[R3] The time improvement is very marginal with $N$ times more memory usage.** As we showed in experiments,
the values of $N$ are typically small and additional memory is not an issue since NODE has a few parameters. Also, we
respectfully disagree that improvement is very marginal.

**[R3] The interpolation acts as an approximation for $z(T)$. Is it possible to directly backpropagate through this
approximated $z(T)$ to obtain gradients with $\mathcal{O}(N)$ memory?** No, it is not possible since $z(T)$ is computed with an
ODE solver without exploiting the interpolation technique.

**[R4] The boundedness condition on $\frac{\partial f}{\partial \theta}$ and $\frac{\partial^2 f}{\partial \theta^2}$ is very strong. Can this even be verified in the considered
applications?** In the considered applications, the norms of $\frac{\partial f}{\partial \theta}$ and $\frac{\partial^2 f}{\partial \theta^2}$ can be computed during training. These norms
roughly estimate the bounds of gradient and hessian of the right-hand side w.r.t. parameters. The potential approach to
ensure the mentioned boundedness is to use proper normalization layers in the right-hand side $f$.

[Meta-Review · NeurIPS 2020]

The four reviewers, all of whom are domain experts, agree that this is a good paper that delivers a delicate but useful methodological contribution to the growing area of NODEs. It should thus be accepted However, the reviewers have also raised several suggestions and requests for improvements. Please make sure to address them as much as possible to ensure this paper reaches its audience.